



# HANZE: a pan-European database of exposure to natural hazards and damaging historical floods since 1870

Dominik Paprotny[1], Oswaldo Morales-Nápoles[1], Sebastiaan N. Jonkman[1]

[1]Department of Hydraulic Engineering, Faculty of Civil Engineering and Geosciences, Delft University of Technology, Stevinweg 1, 2628 CN Delft, The Netherlands

Correspondence to: Dominik Paprotny (d.paprotny@tudelft.nl)

**Abstract.** The influence of social and economic change on the consequences of natural hazards has been a matter of much interest recently. However, there is a lack of comprehensive, high-resolution data on historical changes in land use, population or assets available to study this topic. Here, we present HANZE database, or 'Historical Analysis of Natural Hazards in Europe', which contains two parts: (1) HANZE-Exposure with maps for 37 countries and territories from 1870 to 2020 in 100 m resolution and (2) HANZE-Events, a compilation of past disasters with information on dates, locations and losses, currently limited to floods only. The database was constructed using high-resolution maps of present land use and population, a large compilation of historical statistics, and relatively disaggregation techniques and rule-based land-use reallocation schemes. Data encompassed in HANZE allow to 'normalize' information on losses due to natural hazards by taking into account inflation as well as changes in population, production and wealth. Database of past events currently contains 1564 records (1870–2016) of flash, river, coastal and compound floods. HANZE database is freely available at https://doi.org/10.4121/collection:HANZE

## 1 Introduction

Natural hazards take place when recurring extremes of the Earth's environment collide with human activities. Beyond the natural or anthropogenic changes to the environment, the extent of those activities has profound effect on the consequences of disasters. Even in a space of a few decades, social, economic and technological developments drive the constant evolution of exposure and vulnerability to hazards. Therefore, there is growing interest in how much the number of persons and assets at risk has changed over time worldwide (Jongman et al., 2012, Kummu et al., 2016, Schumacher and Strobi, 2011), and what consequences those findings have for observed trends in natural hazards-related losses (Bouwer, 2011, Bouwer et al., 2007, Daniell et al., 2011, Munich Re, 2016, Schiermeier, 2006).

Floods in Europe have received particular attention (Barredo, 2007). Barredo (2009) found that correcting reported flood losses for inflation and economic growth yields no trend for 1970–2006, in contrast to steep rise in originally-reported losses. Similar findings were presented for the United Kingdom, covering years 1884–2013 (Stevens et al., 2016). Other studies on trends in flood exposure were carried out e. g. for Austria (Fuchs et al., 2015), Italy (Domeneghetti et al., 2015),





the Netherlands (Jongman et al., 2014), Spain (Barredo et al., 2012), Switzerland (Röthlisberger et al., 2016), and the United Kingdom (Stevens et al. 2015). The importance of not only population or economic growth, but also land use distribution has been emphasised (Boudou et al., 2016, Sofia et al., 2017). At the same time, information on past flood losses are being collected in national (Guzzetti, F. and Tonelli, 2004, Haigh et al., 2015) and international databases (Brakenridge, 2017, Guha-Sapir et al., 2017, Munich Re, 2017), including data collected as part of European Union-mandated preliminary flood risk assessments (European Environment Agency, 2015).

Yet, there are several limitations of the aforementioned studies and databases. Exposure datasets were derived at a variety of spatial and temporal resolutions with different thematic coverage. Within a given country, typically one series of population, gross domestic product, housing stock, or other variable were used to normalize reported flood losses. This approach neglects substantial variation in development within countries. Also, the availability of past flood damage information is very uneven between countries and international databases only provide reasonable coverage beginning in the 1980s. The timespan of the studies on exposure is usually limited to the most recent decades, given the lack of adequate data. A commonly-used global dataset of historical population and land use, HYDE (Klein Goldewijk et al., 2010, 2011), has insufficient resolution (5' or 4–7 km over Europe) for use in a flood analysis; it also doesn't contain economic data that could be used to normalize financial losses.

Drawing from recent developments in pan-European demographic and land use mapping, as well as new studies on historical changes in population, production and wealth, we seek to address the aforementioned weaknesses with a new comprehensive dataset. HANZE, or 'Historical Analysis of Natural Hazards in Europe', is a database enabling the study of historical trends and driving factors of vulnerability to natural hazards, with a particular focus on floods. It has two components, namely HANZE-Exposure and HANZE-Events. HANZE-Exposure consists of high-resolution gridded data with information on land use, population, production and wealth per 100 m grid cell from 1870 to 2020. It allows to derive potential damages for any past natural hazard with a defined spatial extent. The other component, HANZE-Events, contains information on location, time and quantitative data on consequences of past natural disasters, currently limited to floods (1870–2016). It is supplemented by economic data necessary for converting nominal monetary losses to a single benchmark. HANZE covers 37 European countries and territories constituting approximately 70% of the continent's population (Eurostat 2017). Detailed composition of the domain can be found in Supplementary File 1 and Supplementary Fig. 1.

As presented in Fig. 1, the starting points for constructing HANZE-Exposure database were a gridded land cover/use map (100 m resolution) and a population map (1 km resolution), both referring to the situation in Europe circa 2011. Based on previously published methods, demographic and economic data were disaggregated to 100 m resolution, and changes in historical land use and population were modelled utilizing a large compilation of historical statistics at the regional level. HANZE-Events was created from a wide array of published sources and databases. The end-date of HANZE-Exposure is different from HANZE-Events, because exposure data are prepared with a 10-year timestep for 1870–1970 and a 5-year for 1970–2020. Therefore, a short-term projection for 2020 is necessary to calculate exposure for post-2015 events. It should be noted that the starting year of 1870 was chosen mainly due to data availability.





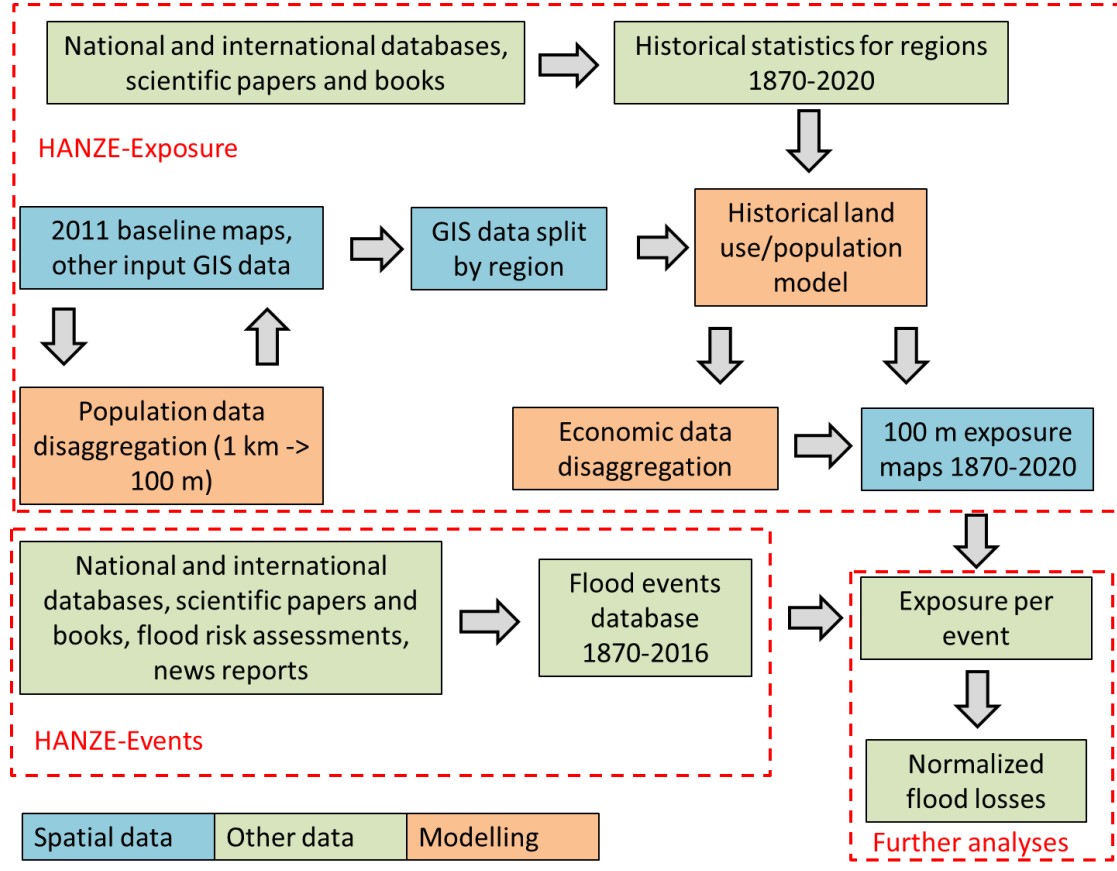

**Figure 1. Workflow in the HANZE database from input data sets to final exposure maps and flood events database and example how the two components interact to derive normalized flood losses.**

## 2 Methods

The creation of HANZE-Exposure data involved four major steps, which are explained below. Main sources and concepts for HANZE-Events are outlined afterwards.

### 2.1 Exposure step 1: baseline maps

There are very few high-resolution population and land cover/use maps, and datasets constructed with a certain methodology rarely extend beyond a single time point. Therefore, two maps (one each for population and land cover) for a single year (2011 or 2012) were collected as baseline for the study. All other time points between 1870 and 2020 are calculated from those baseline maps using historical statistics with substantially lower resolution.



The baseline land cover/use is based on Corine Land Cover (CLC) 2012, version 18.5a (Copernicus Land Monitoring Service 2017). CLC is a project supervised by the European Environment Agency. It has since produced four pan-European land use maps for 1990, 2000, 2006 and 2012. The maps are prepared mostly by manual classification of land cover patches from satellite imagery. For the latest edition, images collected during 2011–2012 were used. The inventory consists of 44 classes (Supplementary Fig. 3). The minimum size of areal features is 25 hectares. For linear objects such as roads, railways, rivers etc., a minimum width of 100 m is used. CLC 2012 covers the entire domain with the exception of Andorra. For this particular country, the land cover/use map was constructed by overlaying data from four different sources, top-to-bottom:

1) CLC 2012 v18.5a, which covers a small strip around the border;

2) CLC 2000 v18.5, an earlier edition which covers a larger strip around the border (Copernicus Land Monitoring Service 2017);

3) Open Street Map, accurate as of mid-2016 (Gisgraphy 2017);

4) Global Land Cover 2000 (Joint Research Centre 2015).

The final map for the full domain of 37 countries and territories is presented in Supplementary Fig. 2.

The baseline population map is based on the GEOSTAT 2011 population grid, version 2.0.1 (Eurostat 2017). This dataset has a 1 km resolution and for most countries it represents the actual population enumerated and georeferenced during the 2011 round of population censuses, complemented by estimates by the European Commission's Joint Research Centre. This dataset is presented in Supplementary Fig. 4. For this study, the 1 km grid had to be further disaggregated to a 100 m resolution. Several methods have been proposed for this procedure and tested for Europe (Gallego, 2010, Gallego et al., 2011). Here, we combine methods M1 and M3 described in Batista e Silva et al. (2013). M1 denotes the 'limiting variable method' used in cartography for creating dasymetric maps of population density. The procedure is an iterative algorithm applied separately for each 1 km grid cell. The steps are as follows:

1) First, uniform population density is assigned for each land use class in a 1 km grid cell:

$$Y_{LG}^0 = Y_G = \frac{X_G}{S_G} \tag{1}$$

where $Y_{LG}^0$ is the population density for land use $L \in \{1, \dots, n\}$ in grid cell $G$ at step 0, $Y_G$ is the population density in the grid cell, i. e. population number $X_G$ divided by area $S_G$.

2) A population density threshold $T_L$ is defined for each one of $n$ land use classes.

3) Land use classes are ranked and the subindex $L$ is renumbered from lowest to highest population density, i.e. $L = 1$ denotes the least densely population land use class in the grid cell

4) Proceeding in order starting with $L = 1$, in step $L$ the density attributed to class $L$ in the previous step is modified if it is above the threshold, i.e. if $Y_{LG}^{L-1} > T_L$. That creates a surplus population $U_{LG}^L$:

5) $\quad U_{LG}^L = S_{LG} \times (Y_{LG}^{L-1} - T_L) \tag{2}$

6) Surplus is then redistributed among the remaining land use classes $M$, hence:



$$Y_{LG}^L = T_L \tag{3}$$

$$Y_{MG}^L = Y_{MG}^{L-1} + \frac{U_{LG}^L}{\sum S_{MG}}, M > L \tag{4}$$

7) If after completing all iterations there is still surplus population, i.e. if $X_G > \sum T_L S_{LG}$, it is redistributed proportionally to the threshold:

$$Y_{LG} = \frac{T_L X_G}{\sum T_L S_{LG}} \tag{5}$$

The crucial aspect of this method is defining the thresholds $T_L$. Here, we use thresholds suggested by Eicher and Brewer (2001), i.e. the 70[th] percentile of the population density of grid cells for which only one land use class was reported in our baseline land use map. Such "pure" cells constituted around 5% of all population grid cells. The final thresholds $T_L$ are shown in Supplementary Table 1. For artificial surfaces other than urban fabric, the CLC classes were merged for the threshold calculation, as very few, if any, "pure" cells could be found for each of those classes. Also, for all areas covered by wetlands, water, sand, glaciers, bare rocks or burnt vegetation the threshold was set at 0, as those terrains are in principle uninhabitable. It should be noted that land use classes with $T_L = 0$ are still included in the algorithm described above.

The result of the calculation, however, is only the population per land use in each 1 km grid cell. Hence, the population had to be disaggregated further. For this we used an approach similar to method M3. This method redistributes the population proportionally to the level of soil sealing, or imperviousness of the ground. This variable has a range from 0%, which indicates completely natural surface, to 100%, which indicates land completely sealed by an artificial surface. This information could not be used directly to redistribute the population as large soil sealing may be caused both by residential and non-residential buildings as well as infrastructure. However, large elements of infrastructure or industry were already taken into account using the 'limiting variable' method.

Data on soil sealing were obtained from the Imperviousness 2012 dataset (Copernicus Land Monitoring Service 2017). It was created based on high-resolution satellite photos taken during 2011-12 in visible and infrared spectrum. This dataset has a 100-meter resolution, which was resampled to a 1 km grid, so that average population density in grid cells with given imperviousness could be calculated. The resulting relationship can be approximated as a power law function, based on cells imperviousness ranging from 1% to 96% (Supplementary Fig. 5). Cells with 0% imperviousness should not be in principle inhabited. Additionally, a power law function converges at 0%. At the opposite on scale, almost no 1 km cells have values above 96%. Hence, the population $X_g$ in 100-meter grid cell $g$ is equal to:

$$X_g = \left[ \frac{Z_g}{\sum Z_g} Y_{LG} S_{LG} \right] \tag{6}$$

where $Z_g$ is the population of grid cell $g$ obtained from the power function divided by the maximum population (at 96% imperviousness):

$$Z_g = \frac{19.479 V_g^{1.3195}}{8031} \tag{7}$$



where $V_g$ is the imperviousness in grid cell $g$. The population $X_g$ is rounded to the closest integer, as population numbers need to be integers. However, rounding can cause difference between the population $X_{LG}$ before and after disaggregation through soil sealing. In such a case, the population is added or subtracted randomly (with equal probability) within the land use class, one person at a time, until the population $X_{LG}$ matches the value before the second stage of disaggregation. This

completes the process, an example of which is shown in Fig. 2.

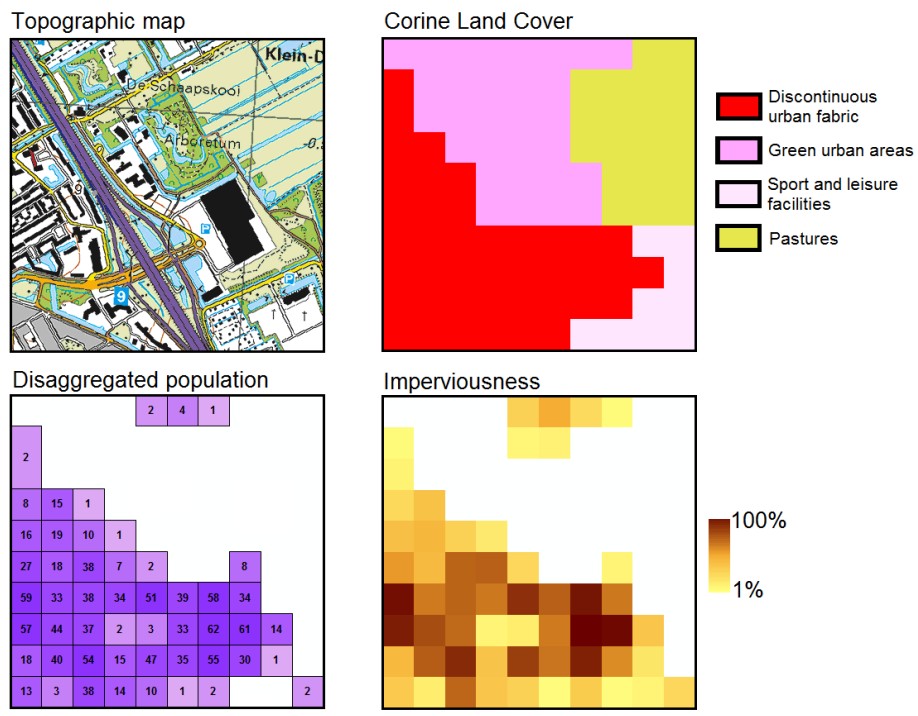

**Figure 2. Disaggregation result and source data for a fragment of the city of Delft in the Netherlands. The area shown corresponds to a 1 km grid in the GEOSTAT population dataset. In this grid cell, the population at the time of the 2011 census was 1218. The**
**top left box was extracted from a Dutch 1:25,000 topographic map for reference. The top right box shows the land use structure according to Corine Land Cover 2012, and the bottom right box shows soil sealing according to Imperviousness 2012 dataset. The final disaggregated 100 m population grid is presented in the bottom left box.**

**2.2 Exposure step 2: historical statistics**

Reconstruction of exposure for years other than the baseline maps requires historical statistics for several variables.
Most of those statistics have been collected at regional level. European Union's Nomenclature of Territorial Units for Statistics (NUTS), 2010 edition (European Union 2011) was used here to define the region. This classification has 4 levels (0, 1, 2, 3), where 0 is the national level and 3 is the finest regional division. Level 3 was chosen for this study, resulting in 1353 regions in the study area (Supplementary Fig. 6). A vector map of regions was obtained from ESRI (2016) with amendments based on Eurostat (2017) map in order to fully match NUTS 2010 classification. Coastlines in the vector map





were further adjusted using the aforementioned CLC 2012 map. NUTS favours administrative divisions in defining the regions, though often statistical (analytical) regions are used instead, created by amalgamating of smaller administrative units. It should be noted that NUTS 2010 was used instead of newer editions because 2011 census data, matching the baseline population map, were disseminated using this classification of regions.

5      All variables collected and used as input to HANZE-Exposure are listed in Table 1. Detailed definitions and concepts for all variables are include in Supplementary File 1. Their utility for the study is explained in the subsequent subsections. In general, all variables were collected from almost 300 sources, so that a time series for one variable for one country was typically merged from several sources. Due to the number of sources and transformations required to complete the database, only the most important methods and sources are mentioned in the Supplementary File 1. Full descriptions of

10     sources and methods are included per country, separately for each variable, with the exception of the 'forestry index', 'airports' and 'reservoirs' variables which are described in this manuscript as they were compiled in a more straightforward manner.

**Table 1. Input variables in HANZE-Exposure**

| Variable | Unit | Resolution |
|---|---|---|
| Total population | Thousands of persons | Regions, 5/10-yearly |
| Urban fraction | Urban population as % of total population | Regions, 5/10-yearly |
| Persons per household | Mean number of persons | Regions, 5/10-yearly |
| Croplands | % of total area | Regions, 5/10-yearly |
| Pastures | % of total area | Regions, 5/10-yearly |
| Infrastructure | Area covered by road and rail infrastructure in ha | Regions, 5/10-yearly |
| GDP | Millions of euros in constant 2011 prices | Regions, 5/10-yearly |
| GDP from agriculture | % of total GDP | Regions, 5/10-yearly |
| GDP from industry | % of total GDP | Regions, 5/10-yearly |
| GDP from services | % of total GDP | Regions, 5/10-yearly |
| Wealth in housing | % of total GDP | Countries, 5/10-yearly |
| Wealth in agriculture | % of GDP from agriculture | Countries, 5/10-yearly |
| Wealth in industry | % of GDP from industry | Countries, 5/10-yearly |
| Wealth in services | % of GDP from services | Countries, 5/10-yearly |
| Wealth in infrastructure | % of total GDP | Countries, 5/10-yearly |
| Forestry index | % of GDP from agriculture | Countries, 2011 |
| Airports | Year of construction | CLC patches, annual |
| Reservoirs | Year of construction | CLC patches, annual |



## 2.3 Exposure step 3: land use and population change modelling

After the baseline maps and a database of historical statistics were completed, changes in land use and population over time were modelled. This was carried out for each of the 1353 NUTS 3 regions separately in specified order. A summary of the procedure is included in Table 2 and the most important details of the methodology are described below.

**Table 2. Summary of historical land use and population modelling approaches, by Corine Land Cover classes (see Supplementary Fig. 3). The number in first column indicates the order in which the modification of land use and population was done.**

| Order | Land use and population type | Modelling approach |
|---|---|---|
| 1 | Urban fabric (CLC 111 & 112) and urban population redistribution | Population per urban grid cell is modified according changes in mean number of persons per household. Surplus population (the difference between urban population in a region after this modification and the value reported in the historical statistics database) and urban fabric is removed starting with grid cells furthest away from urban centers (see text for details) |
| 2 | Industrial or commercial units (CLC 121) | Area of CLC 121 in a region changes proportionately to industrial production per capita in constant prices. 'Industrial' grid cells located furthest from the urban centres are removed first when going back in time |
| 3 | Reservoirs (part of CLC 512) | Reservoirs are removed completely using the information on year of construction. 1069 objects and their construction year were identified using GRanD database (Lehner et al., 2011). |
| 4 | Road and rail networks and associated land (CLC 122) | Area of CLC 122 in a region changes as defined in the historical statistics database. 'Infrastructure' grid cells located furthest from the urban centres are removed first when going back in time |
| 5 | Airports (CLC 124) | Airports are removed completely using the information on year of construction. 1548 objects were identified using mostly OurAirports (2017) database and year of construction was mostly obtained from various language editions of Wikipedia. |
| 6 | Construction sites (CLC 133) | All construction removed from the land use map for years 1870–2005, otherwise as in the baseline map |
| 7 | Croplands (CLC 211–223 & 241–244) | The area covered by croplands in a region is adjusted to match the value in the historical statistics, so that the grid cells least suitable for agriculture are removed first, while unutilized grid cells with the highest suitability are added first. Suitability is proportional to slope and crop suitability index for high-input cereals by FAO (2016). Same-ranked grid cells disambiguated with distance from urban centres (see text for details) |
| 8 | Pastures (CLC 231) | As for croplands, but with crop suitability index for high-input alfalfa used instead of cereals (see text for details) |
| 9 | Burnt areas (CLC 334) | All burnt areas removed from the land use map for years 1870–2000, otherwise as in |





| | | the baseline map |
|----|----|----|
| 10 | Natural areas other than water (CLC 311–333 & 335–422) | If after application of previous steps some land becomes unoccupied, it is assumed that this land was covered by the same natural land cover that is typical in its nearest neighborhood (the most frequently occurring one within 200 m from the outline of the grid cell in question). If no natural land cover was located in the vicinity, the unoccupied land was assumed to be covered by forest (CLC 311). |
| 11 | Rural population redistribution | Population of grid cells which were changed from urban to non-urban is modified, then non-urban population is modified according changes in mean number of persons per household. If needed, rural population added/removed based on distance from urban centres to match historical statistics for a region (see text for details) |
| 12 | Remaining land use (CLC 122, 131, 132, 141, 142, 423, 511, 521–523) | Assumed constant, as in the baseline map |

Redistribution of population within urban areas and growth of cities was modelled based on two factors: change in urban population size and change in number of persons per households. Increasing population combined with smaller families in each dwelling have caused a substantial increase in demand for housing. Between 1870 and 2011, the number of urban households in Europe increased eight-fold. Those extra dwellings were typically constructed outside the urban centres, as existing houses are rarely replaced by bigger ones. Many studies have shown a functional relationship between population density and distance from the city centre (Berry et al., 1963, Anas et al., 1998, Papageorgiou, 2014). Clark (1967) has shown that over time, the sharp decline in population density with distance has become much less pronounced. This is largely caused by the aforementioned social change: in the existing households families become smaller, thus the population declines closer to the centre and the surplus population is accommodated further from the centre in less-developed areas.

In light of the above, the modelling procedure is as follows:

1) In every urban fabric grid cell $g$ in region $r$ the population $P$ in time step $t$ is modified relative to $t-1$ (2011 baseline is step 0) to account for change is household size:

$$P_{t,r,g} = P_{t-1,r,g} \frac{H_{t,r}}{H_{t-1,r}}$$  (8)

where $H_{t,r}$ is the average number of persons per household in each region;

2) All grid cells in a NUTS 3 region are ranked by distance from urban centres, where the highest-ranked cells are the closest to any urban centre.

3) Surplus population $S_t$ is calculated:

$$S_{t,r} = (U_{t-1,r} - U_{t,r})H_{t,r} - U_{t,r}$$  (9)

where $U_{t,r}$ is the urban population in the region according to the historical statistics database;





4) If $S_t$ is positive, it means that the urban area in time step $t$ was smaller relative to $t$-$1$. Urban grid cells are removed starting with the lowest-ranked, and their population is removed as well, until the urban population in the region matches the desired value of $U_{t,r}$.

5) If $S_t$ is negative, it means that the urban area in time step $t$ was larger relative to $t$-$1$. Land use in non-urban grid cells are replaced by CLC 112 class starting with the highest-ranked. In each such grid cell, the population is increased to the threshold value of 65 persons (as defined in Supplementary Table 1), unless it is already higher than that. Urban areas are not allowed to sprawl into uninhabitable areas (Supplementary Table 1).

The important aspect influencing the result of this process is the "distance from urban centre". Urban networks have several levels of hierarchy, with large agglomerations influencing population distribution far outside their borders. Therefore, the distance from urban centre is a weighted sum of three Euclidean distances from:

- Centres of large agglomerations, as presented in a shapefile dataset from United Nations (2014), which shows the arbitrary centres of cities with a population larger than 300,000;

- Centroids of population clusters. Those clusters were calculated by Eurostat (2017) from the 1 km population grid. The centroids were weighted, based on the population in each grid cell;

- Centroids of patches of urban fabric. The patches were taken from Corine Land Cover 2012 (Copernicus Land Monitoring Service 2017), and centroids are based on the geometry of those patches.

Equal weighting of the three layers was found optimal using an analysis of the accuracy of this approach (see Technical Validation section). After urban fabric and population is redistributed, changes in area covered by other types of artificial surfaces, as well as reservoirs, are accounted for (see Table 2). Then, evolution in cropland area is modelled using an approach similar to one utilized in HYDE database of historical land use and population (Klein Goldewijk et al. 2011). It involves changing the allocation of croplands over time according to the land's suitability for agriculture. Therefore, if in time step $t$ the cropland area was smaller than in time step $t$-$1$, 'cropland' grid cells are removed according to their ranking of suitability, starting with the lowest ranked cell (least suitable for croplands), until the value of cropland area in the historical statistics database is achieved. Conversely, if in time step $t$ the cropland area was larger than in time step $t$-$1$, 'non-cropland' grid cells are changed to CLC class 211 (non-irrigated agricultural land) starting with the highest ranked cell.

The suitability is a sum of two indicators, which were also used in the HYDE database. The first indicator is the slope of the terrain (Supplementary Fig. 7), which is a serious limiter to agricultural activity, and which was calculated from EU-DEM dataset at 100 m resolution (Eurostat 2017). We found a close relationship between percentage of area used for croplands and slope, of exponential type. The second indicator is the crop suitability index for high-input cereals as calculated by FAO in the Global Agro-Ecological Zones (GAEZ) database (FAO, 2016, Fischer et al., 2002). The resolution of this dataset is 5' (about 4–7 km, depending on location). The index combines data on climate (1961–90), soil and terrain to estimate potential yield of various crops. Out of several crops tested, high-input cereals (Supplementary Fig. 8) have highest correlation with cropland fraction, of second-order polynomial type.



For the slope indicator, the upper bound was set at 0% slope, while for the crop suitability index the upper bound was set at the polynomial function's maximum (approx. 1500). The suitability indicator for croplands $I_c$ in a given grid cell is thus:

$$I_c = \frac{0.5299e^{-0.063S}}{0.5299} + \frac{-1.6 \cdot 10^{-7}C^2 + 5.6 \cdot 10^{-4}C + 0.143}{0.6327} \qquad (10)$$

5     where $S$ is the slope and $C$ is the crop suitability index.

The main drawback of the method is that due to the relatively coarse resolution of the GAEZ dataset, there are often many cells with the same rank, and the total area of croplands from the model does not exactly match the data in the historical statistics database. Therefore, when too many cells have the same rank, they are further ranked by the centroid distance (as for urban population), so that agricultural land with a given suitability class is added first closer to urban areas, 10   and removed first furthest away from urban areas.

Modelling the changes in pastures follows the same methodology as croplands; only the crop suitability index for cereals was replaced by the same index for high-input alfalfa (also known as lucerne), a common crop growing on meadows and pastures (Supplementary Fig. 9). The suitability indicator for pastures $I_p$ in a given grid cell is thus:

$$I_p = \frac{0.1272e^{-0.047S}}{0.1272} + \frac{-6.9 \cdot 10^{-8}C^2 + 1.7 \cdot 10^{-4}C + 0.0293}{0.1356} \qquad (11)$$

15   The indicators and functional relationships used for analysing agricultural land use changes are included in Supplementary Fig. 10 and 11. After modelling croplands and pastures, burnt areas are removed where necessary (see Table 2) and unoccupied land is replaced by natural vegetation. The final step is thus the redistribution of rural population. The procedure, similar to one employed for urban population, is as follows:

1)   For a given time step $t$ and region $r$, the difference between rural population $R_{t,r}$ in non-urban grid cells (after 20     application of all previous procedures in a given time step) and the rural population according to the NUTS 3 database $N_{t,r}$ was calculated:

$$W_{t,r} = R_{t,r} - N_{t,r} \qquad (12)$$

2)   If $W_{t,r} > 0$, the population of formerly urban grid cells $u$, which transitioned from urban to non-urban during the time step, was modified. Otherwise, this step was omitted. If the population of former urban grid cells was 25     higher than the surplus, i.e. $\sum R_{t,r,u} > W_{t,r}$, the population number in all those cells was reduced by the same proportion, so that the rural population in the region would match the NUTS3 database:

$$R_{t,r,u} = R_{t,r,u} \frac{W_{t,r}}{\sum R_{t,r,u}} \qquad (13)$$

3)   If $W_{t,r} < 0$, the population number in all those cells was reduced to zero, i.e. $R_{t,r,u} = 0$.

4)   Then, the population in all non-urban grid cells was modified according to the change in average household 30     size, i.e.:



$$R_{t,r} = R_{t-1,r} \frac{H_{t,r}}{H_{t-1,r}} \tag{14}$$

where $R_{t,r}$ is the rural population in region $r$ in time step $t$, and $H_{t,r}$ is the average household size.

5) In case there the realized $R_{t,r}$ and expected $N_{t,r}$ number of rural population are still different, population is added or subtracted iteratively, one person at a time to/from a inhabitable, non-urban grid cell (CLC classes 211 to 324, see Supplementary Table 1), starting with those closest to the urban centre, until $R_{t,r} = N_{t,r}$.

## 2.4 Exposure step 4: disaggregated economic data

Disaggregation of economic data provides estimates of GDP and wealth per grid cell, just like the population and land use data. It was carried out after historical gridded population and land use were obtained. The methodology presented here extends the approach proposed in the European Union's ESPON 2013 Programme (Milego and Ramos, 2011) and some others studies, such as G-Econ project (Nordhaus and Xi, 2011), in which the GDP is disaggregated proportionally to the population. This approach works well with a relatively coarse resolutions of the output grid, however at 100 m resolution the economic variables are much less connected with the place of residence of the population. On the other hand, all economic activities still require labour input. Using the observation that employee's compensation constitute approximately half of GDP in European countries (Eurostat 2017), the GDP and wealth are disaggregated in equal proportion using population and land use. It should be noted that wealth is defined here as tangible, produced, non-financial fixed assets. Detailed composition of wealth is described in the Supplementary Information and Supplementary Table 2.

Table 3 provides a summary of the assumptions behind the disaggregation. Additional assumptions had to be made for the agricultural sector, which is the most dispersed, as almost three-quarters of the study area are covered by agricultural land use or forests. At the same time, farmland and pastures are more productive and contain more assets than forests, especially since trees do not count as fixed assets. However, breakdown of GDP by agriculture and forestry is not available at regional level, and very limited historical data exist with such detail on national level. Hence, agricultural GDP and wealth at the regional level were broken down to forestry (including logging) and remaining agriculture (including fishing and aquaculture) using the sectoral split at national level in 2011 from Eurostat (2017). The share of forestry in the agricultural sector varies from zero in Malta to 73% in Sweden.

Half of the GDP generated by agriculture (excluding forestry), as well as half of the wealth in this sector is distributed proportionally to the population living in agricultural areas. The other half was distributed equally among CLC classes 211–244 ("agricultural areas"). GDP and wealth in forestry was distributed the same way, but using CLC classes 311–313 ("forests"). Half of GDP and wealth in industry and services was distributed proportionally to the population in all grid cells, while the other half was distributed equally among specific land use classes where given production is concentrated, as in Table 3.

For the remaining two classes of wealth, the approach was slightly different. The whole wealth in housing (dwellings) was distributed proportionally to the population in all grid cells. The entire value of infrastructure, on the other





hand, was distributed equally over selected land use classes: urban fabric, airports, ports, roads and railway sites (CLC 111, 112 & 122–124).

**Table 3. Disaggregation of economic variables by population and land use classes (CLC = Corine Land Cover).**

| Variable | Category | Disaggregated into population | Disaggregated into land use |
|---|---|---|---|
| GDP | Agriculture excl. forestry | Population in CLC211–244 | CLC211–244 |
| GDP | Forestry | Population in CLC311–313 | CLC311–313 |
| GDP | Industry | Total population | CLC121 |
| GDP | Services | Total population | CLC111–121/133/141/142 |
| Wealth | Housing | Total population | - |
| Wealth | Agriculture excl. forestry | Population in CLC211–244 | CLC211–244 |
| Wealth | Forestry | Population in CLC311–313 | CLC311–313 |
| Wealth | Industry | Total population | CLC121 |
| Wealth | Services | Total population | CLC111–121/133/141/142 |
| Wealth | Infrastructure | - | CLC111/112/122–124 |

**Database of flood events**

HANZE-Events includes information on past damaging floods that occurred in the domain (37 countries and territories) between 1870 and 2016. Several rules were applied to determine whether a flood event indicated in sources should be included in the database, as follows:

- At least one of four statistics (area flooded, persons killed, persons affected, losses) had to be available for a given event. However, if no persons were known to have been killed or missing in the flood, at least one of the other statistics had to be available.

- Insignificant floods, i.e. events which affected only a small part of one region, with no persons killed and affected less than 200 persons, were not included.

- Available information for a given event had to be sufficient in order to assign month, year, country, regions affected, type of flood and general cause of the event. Flood source (river/lake/sea name), detailed information on the cause and day of the event were not required.

- Floods that were caused by insufficient drainage in urban areas not connected with any river system, floods caused entirely by dam failure unrelated with a severe meteorological event, or caused by geophysical phenomena (such as

tsunamis or Icelandic *jökulhlaup* events) were not included.

- Floods events that had impact on more than one country were split per country as long as data were available on per country basis. Otherwise they were presented as one flood event. Also, in case of an event affecting several regions of a country, when the availability of statistics per region is uneven, the event was split accordingly.



Records of flood events were obtained from a large variety of sources (more than 300), including international and national databases, scientific publications and news reports. The source of information is indicated per event in the HANZE-Events dataset. In the majority of cases, entries taken from international databases were cross-checked with other sources and amended as necessary. Databases worth particular mention are EM-DAT (Guha-Sapir, 2017), Dartmouth Flood

Observatory (Brakenridge, 2017), NatCatService (Munich Re, 2017), European Environment Agency's (2015) database of historical information submitted under Floods Directive, as well as national flood databases of France (Lang et al., 2016), Italy (Guzzetti and Tonelli, 2004), Spain( Dirección General de Protección Civil, 2015), United Kingdom (Black and Law, 2004, Haigh et al. 2015), and several national and regional preliminary flood risk assessments.

In order to convert reported losses from various currencies and reference years to a single benchmark, information

on inflation and currencies were collected. Two tables were prepared and are included with other HANZE input data. The first one includes all currencies that were used in the study area between 1870 and present, with their names, ISO 4217 codes, starting and ending dates of validity as well as conversion factors to euro. For countries not currently using the euro, 2011 exchange rates from Eurostat (2017) were used. Information on currencies and conversion factors was mostly gathered from ISO 4217 (ISO, 2017) and GHOC databases (Taylor, 2004), supplemented by various Internet resources.

The second table contains deflators used to adjust nominal losses to real losses in 2011 prices. The GDP deflator was generally used, as it allowed to make the loss adjustments consistent with GDP values. Only if the GDP was not available, alternative price indices were used, but always 'anchored' to the GDP deflator series. These other series included indices of consumer, wholesale, retail, or cost-of-living prices. The source of the data was usually the same as those for the GDP data; they are listed in detail in the data files themselves. It should be noted that the currency conversions and deflators

omit four cases of hyperinflation: Germany 1923, Poland 1923, Greece 1944 and Hungary 1946. Inclusion of those cases would cause large distortions to the data series. Hyperinflation periods and resulting currency changes were marked in the dataset. The dataset also includes deflator series for three former countries: Czechoslovakia, the Soviet Union and Yugoslavia, as many countries were part of them in the past.

## 3 Results

### 25 3.1 Database contents

The complete list of files of HANZE and their contents is listed in Table 4. Exposure maps in 100 m resolution are provided as GeoTIFF rasters in ETRS89/LAEA projection, consistent with INSPIRE European grid. The baseline maps of land use and population (100 m resolution) are also included. For the benefit of climate research groups in particular, the datasets are provided also in aggregated, lower-resolution versions. Two files in netCDF format are included: 5' grid in

geographical coordinates (WGS84) and finally 0.11° rotated-pole grid as used in EURO-CORDEX climate modelling framework (Jacob et al., 2014).



Input historical statistics as well as the HANZE-Events database of past damaging floods are provided as Excel files. Structure of tables with input data is detailed in Supplementary Tables 3 and 4. Apart from the statistical information, each of the two files (with demographic/environmental and economic data) include a table with all sources and transformations made to the data per country, per variable and finally per year, and a list of references. The contents of the

5 HANZE-Events database with explanations of all data recorded per event is shown in Table 5.

**Table 4. List of files of HANZE database. XXXX = value indicating the year to which dataset pertains.**

| Type | File | Format | Variables / contents |
|---|---|---|---|
| Output | CLC_XXXX | 8-bit TIFF | Land cover/use type, 44 classes according to Corine Land Cover |
| Output | Pop_XXXX | 16-bit TIFF | Total population per grid cell (in persons) |
| Output | GDP_XXXX | 16-bit TIFF | Gross domestic product (GDP) per grid cell per year (x 10,000 euro in constant 2011 prices) |
| Output | FA_XXXX | 16-bit TIFF | Wealth per grid cell (x 100,000 euro in constant 2011 prices) |
| Output | Exposure_5min | netCDF | Land use (fraction of: urban areas, croplands, pastures, forests, water), total population, GDP and wealth per grid cell aggregated to 5 arc minute resolution |
| Output | Exposure_cordex_0.11 | netCDF | As above, but aggregated to EURO-CORDEX rotated-pole grid, 0.11° resolution. |
| Output | Events_floods | Excel | List of past damaging floods (list of variables in Table 5) |
| Input | Expo_input_CLC_Pop | Excel | Historical land use/cover and population statistics |
| Input | Expo_input Econ | Excel | Historical economic and currency statistics |
| Input | CLC_base | 8-bit TIFF | Baseline land cover/use type, 44 classes according to Corine Land Cover |
| Input | Pop_base | 16-bit TIFF | Total baseline (disaggregated) population per 100 m grid cell (in persons) |

**Table 5. Information included in HANZE-Events database.**

| Variable | Description |
|---|---|
| No. | Event number |
| Country code | NUTS0 country code |
| Year | Year of the event (assigned from starting date) |
| Country name | Country in which the event occurred, using political divisions of the time of the event. In case of former countries of Czechoslovakia, East Germany, USSR and Yugoslavia, the appropriate succession state was used instead of the original country |
| Start date | Date on which the flood event started and ended; the exact daily dates are not always known, or are imprecise, but an event was included in the database as long as the starting month could be identified |
| End date | Date on which the flood event ended |
| Type | Type of flood event, which can be River, Coastal, River/Coastal, or Flash. The events were assigned to River/Coastal type if both factors contributed to the flooding. Flash flood type was assigned if the event was |



| | |
|---|---|
| | caused by rainfall lasting less than a day. However, often the information on meteorological conditions was missing and hence division of events into River and Flash floods was made based on dates of the event, location, season and impacts |
| Flood source | Name of the river, lake or sea from which the flood originated, if available. The list of names is usually not comprehensive |
| Regions affected | Regions were flood damages were reported, using the NUTS3 delimitation of regions |
| Area flooded | Area inundated by the flood in $km^2$. This statistic more often than not relates only to agricultural land |
| Persons killed | Number of deaths due to the flood, including missing persons |
| Persons affected | Number of people whose houses were flooded. However, the reported numbers of persons affected often only show the number of evacuees or persons rendered homeless by the event. If no other number was available, those ones were used. If only the number of houses flooded was reported, the number persons affected was estimated by multiplying the number of houses by 4 |
| Losses (nominal value) | Damages in monetary terms, in the currency and prices of the year of the flood event |
| Losses (mln EUR, 2011) | Damages in monetary terms converted to euro, correcting for price inflation relative to 2011 |
| Cause | The meteorological causes of the event, including precipitation values, surge heights, etc. if available |
| Notes | Other relevant information, including co-occurrence of related events such as landslides or dam breaks, information on large discrepancies in the sources, estimated return periods and other relevant statistics |
| Sources | List of publications and databases from which the information was obtained |

### 3.2 Data validation

The accuracy of the data involved in HANZE database is influenced by three elements: (1) quality of baseline maps and historical statistics; (2) robustness of the methodologies used for disaggregation of data and modelling change in population and land use; and (3) completeness and reliability of the records of past damaging floods.

### 3.2.1 Baseline maps and historical statistics

The baseline land cover/use map, Corine Land Cover 2012, was employed for this analysis before final validation was made, but subsequently the map was found to have a thematic accuracy of around 90% (Copernicus Land Monitoring Service, 2017). Still, the use of thresholds of minimum size (25 ha) and width (100 m) of objects necessary for inclusion in the map cause many small objects with high effect on population distribution be omitted, e.g. small bodies of water or smaller pieces of infrastructure and villages. It should be also noted that mapping was done by each country independently, and therefore the classification of land use is not always fully consistent between countries and the thematical accuracy





varied from 82% to 97% between countries. Validation reports are also available for imperviousness layer and elevation models from Copernicus Land Monitoring Service (2017).

The baseline GEOSTAT population grid's accuracy is described in reports by the provider (Eurostat 2017). Though for most countries the quality of the 1 km grid is very high, with 98–100% of national populations georeferenced, there are

exceptions. In Bulgaria, for example, only 57% of the population was georeferenced and the remainder was disaggregated from settlements or local administrative units. In Italy the whole dataset was calculated from enumeration areas, albeit their average size was below 1 km$^2$. For some smaller countries, the population distribution was calculated by the European Commission's Joint Research Centre using land use data. Basic information on GEOSTAT accuracy per country has been included in the HANZE database.

Historical statistics were compiled from a large variety of sources. Total population figures were mostly available at regional level, while the remaining statistics were usually available only at national level beyond the most recent two-three decades. Inevitably, there are inaccuracies from applying national trends at the regional level. Also, economic data series before approx. 1950 for Western Europe and 1990 for Central Europe are more often than not reconstructions based on ancillary or proxy data. Notwithstanding those limitations, we believe that the HANZE database represents an improvement

in resolution and thematic coverage compared with the HYDE database over the study area. A comparison in the number of regional estimates of total and urban population included in both databases is shown in Supplementary Fig. 12.

### 3.2.2 Methods

In this study, the population distribution was disaggregated from 1 km to 100 m using two methods validated previously in literature (Batista e Silva et al., 2013). Lack of comparative data at such resolution prevents us from further analysing the

quality of the disaggregation. Still, the original resolution is very fine and the refining narrows the distribution of population by eliminating areas that are uninhabitable or very unlikely to be inhabited. There is no comparative information for economic variables downscaled from regional level to gridded data.

Lack of comparable data for validation is also evident for historical land use changes. Some local reconstructions of past land cover/use were made from old maps, but there is limited consistency in classification or minimal mapping units to

allow accurate comparison. Corine Land Cover is available for 2000 and 2006, but often indicated changes in land use are only reclassifications rather than actual developments. Hence, changes in historical croplands and pasture distribution were not validated directly. The general methodology used here, i.e. reallocating croplands and pastures based on land suitability for agriculture has been extensively utilized in many studies before (Hurtt et al., 2011, Kaplan et al., 2011, Klein Goldewijk and Verburg, 2013, Pongratz et al., 2008, Ramankutty and Foley, 1999). A more detailed uncertainty and sensitivity analysis

of the input data and methods would be possible using structured expert judgment (Colson and Cooke, 2017, Cooke and Goossens, 2008).





Some analysis, though, could be made on the historical distribution on urban population. Estimates of Clark's model of urban population density are available for 19 cities, which consider population distribution in urban areas as an exponential function (Clark, 1951):

$$y = Ae^{-bx} \tag{15}$$

where $y$ is the population density (in persons per km$^2$), $x$ is the distance from the city centre (in km), $A$ and $b$ are exponential function coefficients. A total of 42 estimates of this equation spanning a whole century, from 1871 to 1971 were collected, of which a complete list can be found in Supplementary Table 5 (Clark, 1951, 1967, Hourihan, 1982). In the population map constructed herein, the population density was calculated for 500 m wide zones around (arbitrarily chosen) city centre, interpolated to match the time points from literature and then fitted to an exponential function.

A comparison of function parameters is presented in Fig. 3. Overall, a reasonable fit was achieved. For cities for which more than one year of data was available, a decline of both parameters over time was observed, as in the literature case studies. A better match of modelled and observed estimates of eq. 13 parameters would be difficult, since the exponential curve fits are very sensitive to the sample size (distance from the city centre) and the source material: literature studies used census wards of different sizes instead of a disaggregated population grid used here.

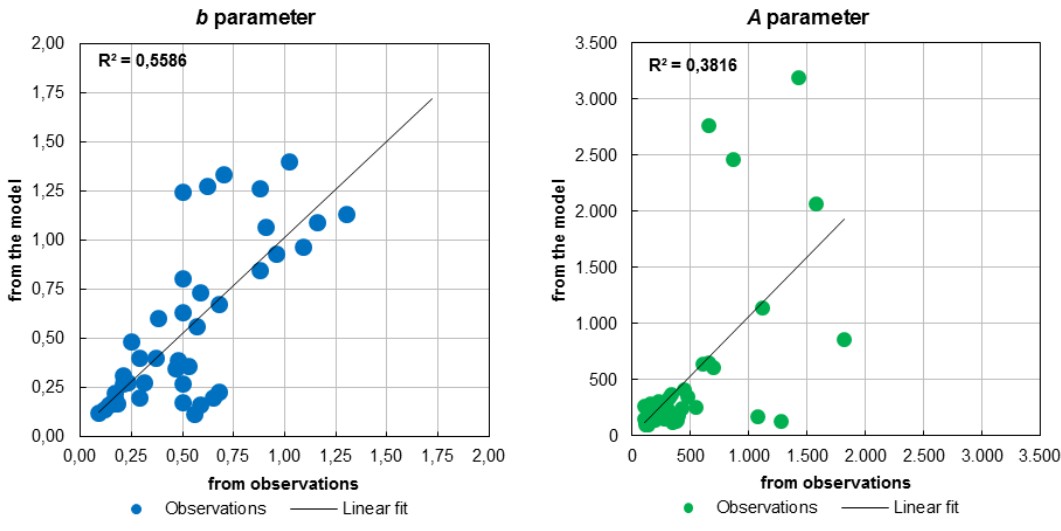

**Figure 3. Estimates of *A* and *b* parameters (eq. 10) from modelled and observed population data.**

### 3.2.3 Records of past damaging floods

The quality of records of past floods depends on two main factors: (1) completeness (what share of past floods could be traced) and (2) the reliability of information on the location and quantitative data on losses. Completeness varies substantially between countries, few of which maintain publicly-available databases of flood losses. Historical information contained in mandatory preliminary flood risk assessments ranged was sometimes very extensive, but often little or no





quantitative information on losses were included. International databases of events have short timespan: EM-DAT nominally starts with year 1900, but very few floods are included before 1970. NatCatService and EEA's compilation of Floods Directive data have coverage from 1980 and Dartmouth Flood Observatory from 1985. Due to the development of Internet, availability of news reports on floods increases substantially starting with mid-1990s, though an increasing number of old

newspaper articles are digitized and provide a valuable resource. Underreporting for Central-European countries before 1990 is also noticeable and is caused by communist-era censorship.

The reliability of past flood loss data remains an open question. Efforts were made to gather multiple sources for past events, especially large ones. In the vast majority of cases, records of floods from international databases can be corroborated by other sources or at least by other international databases. Some records were found to be either dubious or

were not primarily flood events, but rather landslides, as found before e.g. for Portugal (Zêzere et al., 2014) . The most extreme case is a record in EM-DAT, according to which a flood along the Danube in Romania in 1926 caused 1000 deaths. However, the Romanian preliminary flood risk assessment indicates that national literature sources do not contain any mention of flood fatalities in that year (Administrația Națională Apele Române, 2009). A calamity of such magnitude—it would have been the deadliest European flood in the past 150 years—must have left a trace in several sources. Therefore,

this event was not included in HANZE. Also, there are some cases of floods occurring with other hazards (windstorms, hail, landslides), where it was not possible to disentangle flood losses from those from other causes. Therefore some flood records include or might include those other losses, which is marked in the database under "Notes" category. On the other hand, some flash floods were not included if the majority of losses were not caused by floodwater.

In total, HANZE-Events contains 1564 records of floods (Fig. 4), where information on the flooded area are

available for 157 events (10%), persons killed for 1547 (99%), persons affected for 682 (44%) and monetary losses for 560 (36%). The known flood consequences amount to almost 123,000 km$^2$ of land inundated, 18,319 fatalities, 7.5 million people affected and 227 bln euro of damages in 2011 prices. This can be considered only a foundation of a comprehensive database. In future work, many more sources of information could be integrated with a larger pan-European effort to overcome the problem of language barriers and need of physical access to many older sources.





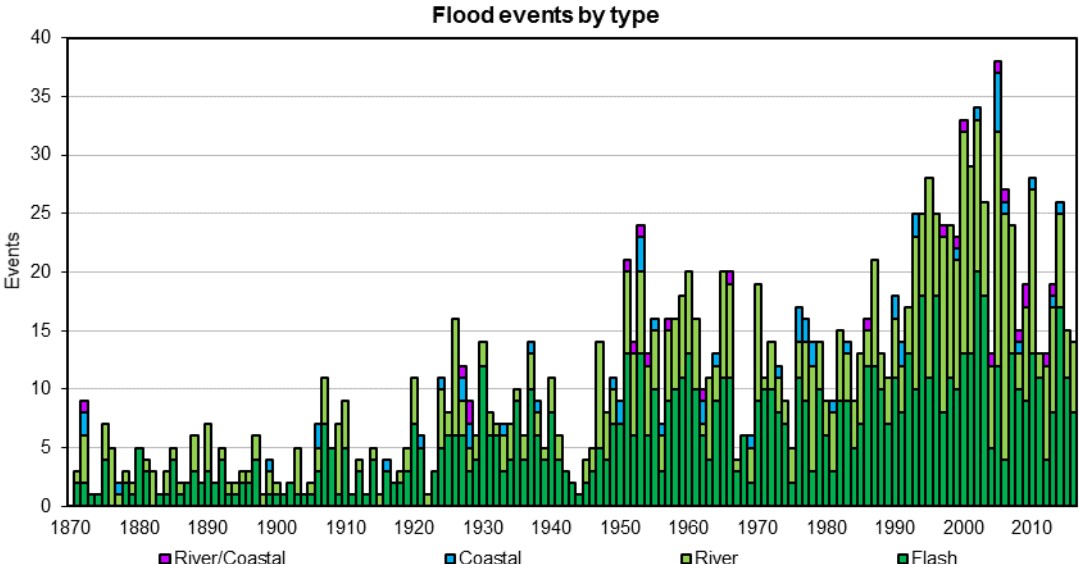

**Figure 4. Distribution of flood events in HANZE by year and type.**

**Data availability**

HANZE-Exposure (both input and output datasets), HANZE-Events and database documentation were uploaded to the 4TU Centre for Research Data (https://doi.org/10.4121/collection:HANZE). Contents of the database was described in section 3.1.

**Conclusions**

The HANZE-Exposure database is intended to provide data allowing to normalize historical losses related to natural hazards. We hope that it will be useful for researchers studying past occurrences of damaging meteorological, hydrological or geophysical phenomena. Also, the database could be used to analyse changes in distribution of population and assets using zones of different level of hazard, e.g. flood hazard maps. To improve reusability, we provide exposure data in different resolutions and formats, so that the dataset can be easily applied regardless of how the extent of events ('footprint') is defined: a polygon, a raster layer, a country subdivision or a climate model grid. HANZE-Exposure can be also considered as a refinement of the HYDE database for Europe for the past 150 years. In principle the spatial datasets and the input historical statistics could also be applied to purely socio-economic research, e.g. studying regional development or land-use changes. However, in that case we would urge potential users to first analyse the methodology and data sources contained in the database and its documentation in order to assess if HANZE-Exposure is suited for the users' research purposes.

HANZE-Events currently encompasses only information on floods, but the same framework could be used for other hazards. Number of casualties or losses in monetary terms can then be corrected (or 'normalized') for changes in currency, inflation, population or economic growth using HANZE-Exposure. Also, reported losses could be contrasted with potential losses, e.g. exposed population or assets within a flood hazard zone with a given probability of occurrence (Paprotny et al., 2017). Information on relative losses could provide insight how vulnerability of population changed over time.

**Author contributions**

D.P. conceived and designed the study, prepared the datasets and drafted a first version of the manuscript. S.N.J. and O.M.N. helped to design the study. O.M.N. helped to draft the manuscript. All authors revised the manuscript and gave final approval for publication.

**Competing interests**

The authors declare that they have no conflict of interest.

**Acknowledgements**

HANZE database was prepared with the support of project "Risk Analysis of Infrastructure Networks in response to extreme weather" (RAIN), which received funding from the European Union's Seventh Framework Programme for research, technological development and demonstration under grant agreement no. 608166. Further support was provided by project "Bridging the Gap for Innovations in Disaster Resilience" (BRIGAID), which received funding from the European Union's Horizon 2020 research and innovation programme under grant agreement no. 700699. The authors would like to thank Antonia Sebastian for her comments on the manuscript.

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
