# Peer review of "HANZE: a pan-European database of exposure to natural hazards and damaging historical floods since 1870"

_Earth System Science Data, 2017_

## Referee Comment (RC1) · Anonymous Referee #1 · 9 Dec 2017

Presented for the review is the dataset by D. Paprotny et al. "HANZE: a pan-European database of exposure to natural hazards and damaging historical floods since 1870", containing a variety of socio-economic data referred to natural hazards, namely floods of different origin, for the majority of European countries. The dataset is provided in both table (xslx) and gridded (multiple available) formats, well documented and easy to access. The concept behind the dataset compilation is a variety of spatial disaggregation methodologies employed to combine multiple heterogenous datasets into comprehensive and consistent database, for provisional use to assess the socio-economic impact from floods. The authors show a significant effort to produce the presented dataset, as the initial data for it should have been thoroughly obtained, analyzed, re-

gridded and remodeled. As it can be seen from the well-referenced corresponding section, the regridding methodology itself is not new, but the authors employ it with full competence in order to create new datasets of land use, total population, gross domestic product and wealth in the research domain. The database of past flood damages for the period 1870 – 2016 is itself an outstanding effort in compiling a large dataset of heterogeneous socio-economic data in a sublime and useful form of an Excel spreadsheet. The dataset is definitely useful for a large group of specialists ranging from hydrologists to socio-economists. The data quality is appropriate for further scientific usage, as well as its presentation. The article is fairly longer than expected, but offers the presentation of concepts behind the dataset creation in detail. However, the authors emphasize that the methodology and data sources of the database should be analyzed by potential users to determine the relevance for their task. I suggest the HANZE dataset is published as is, with the article undergoing minor technical corrections. Noticed typos:

p. 4 line 29: "populated"

p. 8 table 2 row 1 "according to changes"

p. 9 line 13 "change in household"

---

## Referee Comment (RC2) · Anonymous Referee #2 · 9 Jan 2018

General comment

One of the most challenging tasks of Pan-European scale studies on natural hazards, their spatial and temporal dynamics, is to break through the national borders, as all the data is commonly collected and systematized within this borders separately for each country. The presented dataset HANZE collects, systematizes and unifies national datasets on different land use, economical etc. characteristics and historical flood events in one spatially referenced Pan-European database, changes in land use and population over time were modelled. HANZE can be helpful for a wide range of researches. The dataset content and underlying methods of calculation are clearly

described. All files are accessible for download and well-documented. The presented dataset is undoubtedly of interest for the scientific society and would be a good contribution to the journal. However, there are corrections to be made.

Specific comments

The introduction lacks an overview on existing datasets and key difference from the suggested database. Authors address them as a whole mentioning disadvantages, and give only one example (HYDE).

The baseline land cover/use map is of 100 m resolution according to the introduction, while in the "Methods" section it is mentioned that this scale is used only for linear objects, the minimum size of areal features is 25 hectares.

As most of the data has a far less scale (1 km for population, economics and historical statistics for NUTS level 3 regions) than the resulting maps resolution (100 m) a future user of the dataset could face a danger of a false granularity. The authors should comment on the decision to actually model extremely detailed data rather than use a scale compared with the input data scale. Furthermore, the article lacks validation of the disaggregated data. The authors claim a "lack of comparative data", though it seems possible, for example, to compare modelled population density in the sum of cells corresponding to a locality with statistical historical data on this locality etc. Absence of a sufficient validation substantially drops the value of the presented gridded maps and confidence in methods.

The graph for the A parameter (figure 3) doesn't seem a "reasonable fit". The authors should than explain their understanding of a reasonable fit.

The description of the dataset content on https://data.4tu.nl/repository/collection:HANZE is different from the Table 4 that makes it a bit confusing to match one with another.

Technical corrections

Paragraph 30 page 2: "Based on previously published methods. . ." a reference needed.

Figure 2 has no legend for "Disaggregated population". No number for the "Database of flood events" section. Paragraph 5 page 14: a misprint "Spain( Dirección".

---

## Author Comment (AC1) · 25 Jan 2018

We would like thank the referees for the time spent in reviewing our article and their valuable comments. Below, we list all the comments ("C") and our responses ("R").

Reviewer #1

C: [T]he authors emphasize that the methodology and data sources of the database should be analyzed by potential users to determine the relevance for their task.

R: Analyzing how the dataset was constructed is always beneficial, but this particular comment was written with the purpose to highlight the importance of specific research questions that could benefit from our database. For example analysing the convergence of the levels of economic development between regions. In this case, the resolution of regional economic data is of crucial importance, much more than in exposure-normalization. We will make this more clear in the paper.

C: Noticed typos: p. 4 line 29: "populated", p. 8 table 2 row 1 "according to changes", p. 9 line 13 "change in household"

R: The typos will be corrected in the revision.

Reviewer #2

C: The introduction lacks an overview on existing datasets and key difference from the suggested database. Authors address them as a whole mentioning disadvantages, and give only one example (HYDE).

R: A new paragraph will be added to the introduction to give an overview of existing datasets of historical population, land use and GDP.

C: The baseline land cover/use map is of 100 m resolution according to the introduction, while in the "Methods" section it is mentioned that this scale is used only for linear objects, the minimum size of areal features is 25 hectares.

R: The values of 25 ha and 100 m mentioned in section 3.2.1. are the so-called "minimum mapping unit", i.e. the smallest size of objects that are distinguished in a land cover/use map. This should not be confused with the accuracy of representation of objects; CLC is actually generated from satellite images with a resolution of 25 m or better. The vector product is the latter transformed into a raster with a 100 m grid. We will modify the text to make this point clear, as the description is indeed misleading.

C: As most of the data has a far less scale (1 km for population, economics and historical statistics for NUTS level 3 regions) than the resulting maps resolution (100 m) a future user of the dataset could face a danger of a false granularity. The authors should comment on the decision to actually model extremely detailed data rather than use a

scale compared with the input data scale.

R: The 100 m resolution was chosen for two particular reasons. Firstly, the Corine Land Cover data is provided in 100 m resolution. For this reason it was desirable to keep this resolution when reallocating land cover/use in past time-steps, rather than resampling it to a lower resolution and possibly introduce significant errors in losing small, but valuable elements such as urban areas and pieces of infrastructure. Secondly, the 100 m was useful for combining the results with pan-European flood maps, which have the same resolution. Overall, the very fine resolution was most optimal to represent the areas where most population lives and most assets are accumulated: urban areas, industry and infrastructure, all of which cover only a small fraction of Europe. We will make this point clear in the conclusions. Furthermore, the article lacks validation of the disaggregated data. The authors claim a "lack of comparative data", though it seems possible, for example, to compare modelled population density in the sum of cells corresponding to a locality with statistical historical data on this locality etc. Absence of a sufficient validation substantially drops the value of the presented gridded maps and confidence in methods.

R: Looking again at available data, we devised a validation of gridded population estimates based on a dataset of local administrative unit (LAU) level from Eurostat covering years 1960-2010 and a map of communes in Europe. In this way we were able to analyse the accuracy of the population grid for those years. The method of this analysis is described in an attachment to this response to reviews, together with an Excel file with validation results for all NUTS3 regions. He hope that this will allow the users to have a more detailed insight into the dataset's value.

C: The graph for the A parameter (figure 3) doesn't seem a "reasonable fit". The authors should than explain their understanding of a reasonable fit.

R: The statement in the paper is indeed too positive in tone, therefore we will clarify it as follows: "Overall, a good fit was achieved for b parameter, but only a relatively poor

for A parameter." The exponential assumption is in itself an assumption that can be relaxed with further research, as we explore possibilities for other ways of modelling changes in urban population density and validating the population map

C: The description of the dataset content on https://data.4tu.nl/repository/collection:HANZE is different from the Table 4 that makes it a bit confusing to match one with another.

R: This point is not exactly clear to us as to what is the extent of the mismatch. Table 3 of the online documentation is fully aligned to Table 4 of the paper, except only the description of netCDF files in the repository erroneously omits the fact that they also include data on land use. The table in the paper is correct in this manner. We will try to correct the mistake in the repository document.

C: Technical corrections: Paragraph 30 page 2: "Based on previously published methods: : :" a reference needed. Figure 2 has no legend for "Disaggregated population". No number for the "Database of flood events" section. Paragraph 5 page 14: a misprint "Spain( Dirección".

R: The typos will be corrected in the revision and the missing legend will be added to Figure 2.

Please also note the supplement to this comment:
https://www.earth-syst-sci-data-discuss.net/essd-2017-105/essd-2017-105-AC1-supplement.zip